# An Energy-Efficient LoRa Multi-Hop Protocol through Preamble Sampling for Remote Sensing

**DOI:** 10.3390/s23114994

**Published:** 2023-05-23

**Authors:** Guus Leenders, Gilles Callebaut, Geoffrey Ottoy, Liesbet Van der Perre, Lieven De Strycker

**Affiliations:** Dramco, ESAT-WaveCore, KU Leuven, 9000 Ghent, Belgium

**Keywords:** energy efficiency, IoT, LoRa, multi-hop

## Abstract

Internet of Things technologies open up new applications for remote monitoring of forests, fields, etc. These networks require autonomous operation: combining ultra-long-range connectivity with low energy consumption. While typical low-power wide-area networks offer long-range characteristics, they fall short in providing coverage for environmental tracking in ultra-remote areas spanning hundreds of square kilometers. This paper presents a multi-hop protocol to extend the sensor’s range, whilst still enabling low-power operation: maximizing sleep time by employing prolonged preamble sampling, and minimizing the transmit energy per actual payload bit through forwarded data aggregation. Real-life experiments, as well as large-scale simulations, prove the capabilities of the proposed multi-hop network protocol. By employing prolonged preamble sampling a node’s lifespan can be increased to up to 4 years when transmitting packages every 6 h, a significant improvement compared to only 2 days when continuously listening for incoming packages. By aggregating forwarded data, a node is able to further reduce its energy consumption by up to 61%. The reliability of the network is proven: 90% of nodes achieve a packet delivery ratio of at least 70%. The employed hardware platform, network protocol stack and simulation framework for optimization are released in open access.

## 1. Multi-Hop Wireless Networks

### 1.1. Introduction

One of the key areas where Internet of Things (IoT) is currently seeing increased adoption is large-scale, remote monitoring applications. The ability to connect a large number of devices spread out over vast areas can provide significant benefits. In these remote applications, IoT technologies can be used in a wireless sensor network (WSN) to register environmental phenomena: from soil and snow monitoring across large nature reserves [1], to plant stress monitoring across large agricultural fields [2]. These use cases typically require only an extremely small throughput rate and are not latency critical. To maximize the lifespan of the IoT nodes, which are expected to operate on a limited energy supply for several years, it is crucial to optimize both their hardware and software for minimal power consumption [3].

Recently, a few wireless Internet of Things (IoT) technologies have proven their worth in actual deployments. The Long-range wide-area network (LoRaWAN) implementation in particular has become widespread. LoRaWAN is designed specifically for low-throughput communication and offers long-range, low-power, and unlicensed spectrum operation. In these networks, a gateway is connected to the cloud using a single-hop star network topology, as is typical for most IoT networks. Despite the 2–5 km range of coverage that LoRaWAN can accomplish [4], it might not be sufficient to cover the hundreds of square kilometers required in the aforementioned use scenarios, or when propagation is severely attenuated, e.g., because of obstructed paths. Coverage could be provided by distributing a large amount of long-range wide-area network (LoRaWAN) gateways in the area; however, this proves to be unfeasible due to lacking (cellular) network coverage, energy restrictions, or financial cost. Therefore, we propose a long range (LoRa)-based multi-hop network to extend the coverage, specifically optimized for energy efficiency.

A multi-hop protocol allows messages to be transmitted from one node to another through a series of intermediate nodes, effectively allowing packets to *hop* from one node to the next until they reach their destination (see Figure 1). This enables the network to cover a much larger area than would be possible with a single hop. In the implementation presented in this work, we assume only one possible destination: the gateway or the sink, which can store the gathered data or relay them to another cloud-connected network.

Designing an energy-optimized network with multi-hop capabilities comes with a significant challenge, as each node must constantly listen for network activity to forward data. This can result in high energy consumption. However, achieving low-power operation is essential in remote Internet of Things (IoT) nodes to ensure prolonged battery life, reduce maintenance needs, and enable efficient deployment in resource-constrained environments. To check for network activity and enable low-power operation, channel activity detection (CAD) can be used [5]. Receiving messages in the implemented multi-hop network is thus entirely based on a periodic long range (LoRa) CAD mechanism.

### 1.2. Related Work

Multi-hop long range (LoRa) networks have been discussed in the literature. Yet, few authors have focused on energy-efficient tailored network design. Hereby, we summarize related works with respect to the proposed multi-hop network. An overview is presented in Table 1, which is further discussed below.

The comparison between single-hops and multi-hop networks is presented in [6]. The authors determine at what range a multi-hop network is needed, and whether the latter is more energy efficient when considering the packet delivery ratio (PDR) for various distances, spreading factors (SFs), and transmit powers. They show that, to achieve a 90% PDR, a two-hop network can provide up to 50% energy reduction as compared to a single-hop network, while increasing 35% coverage at a particular SF.

**Table 1 sensors-23-04994-t001:** Overview of related work, including the multiple-access technique used, development focus area, and further details of the validation of the protocol in question. ✓, or ✗ indicates whether the condition holds true or not, respectively. ‘-’ indicates not applicable, and ‘?’ indicates not known.

Ref.	Multiple-Access Technique	Focus	Experimentally Validated	Firmware		Validation by Simulations	Simulator	
				Open-Source	Platform		Open-Source	Lang.
**Asynchronous multi-hop Protocols**								
This work	Prolonged preamble sampling	Energy efficiency	✓	✓	Arduino	✓	✓	Python
[7]	Delay tolerant and remote	Localization	✓	✗	Arduino	✗	-	-
[8]	ALOHA SF differentiation	Energy efficiency	✓	✗	LoRaMote	✗	-	-
[9]	ALOHA	Coverage	✓	✗	Mbed	✗	-	-
[10]	FDMA on long range (LoRa) channels	Coverage, throughput	✓	✓	Arduino	✓	✓	Python
[11]	TDOA, SF differentiation	Reliability	✗	-	-	✓	✗	OMNeT++
[12]	ALOHA, SF differentiation	Reliability, latency	✓	✗	STM32	✓	✗	?
**Time-synchronous multi-hop Protocols**								
[13]	On-demand TDMA	Energy efficiency, latency	✓	✗	PIC	✗	-	-
[14]	TSCH	Coverage	✓	✗	LoRaMote	✗	-	-
[15]	Predictive wake-up	Coverage	✗	✗	STM32	✓	-	-
[16]	Predictive wake-up	Coverage	✓	✗	STM32	✗	-	-
[17]	Predictive wake-up	Energy efficiency, coverage	✗	-	-	✓	✗	OMNeT++
[18]	TSCH	Latency	✓	✗	Mbed	✗	-	-
[19]	TDMA	Energy efficiency	✓	✗	Zolertia	✗	-	-
[20,21]	2-hop slot scheduling TDMA	Coverage, reliability	✓	✗	STM32	✓	✗	?

#### 1.2.1. Synchronous versus Asynchronous Networks

To support any multi-hop network technology, some form of multiple access regulation is required. In general, two approaches can be taken with respect to multiple access: time-synchronous or -asynchronous networks [22].

Time-synchronized networks transmit time information to all nodes in the wireless sensor network (WSN). These protocols often utilize variants of time division multiple access (TDMA), where different time slots can be used by different Internet of Things (IoT) nodes. Common periods such as receive/sleep windows can also be defined. As all nodes are time-synchronous, the number of collisions and idle listening time are drastically reduced, yielding an efficient and reliable way of communicating. Yet, setting up and maintaining time synchronization across a vast number of IoT nodes can prove to be challenging, as clock drifts require some sort of recurring time synchronization methods. Hence, as illustrated in [22], the energy cost of keeping the network synchronized is only permissible in high- or medium-load traffic scenarios. As our work focuses on extremely low throughput networks (as discussed in Section 1.1), time-synchronous networks are not applicable and, as such, will not be discussed further.

In asynchronous networks, devices do not operate on any schedule or synchronization signal to coordinate their transmissions. In these networks, each device operates independently and starts transmission at an entirely random moment in time. This method can lead to increased collisions and idle listening. Yet, no time synchronisation needs to be maintained across the network. This makes this type of network ideal for low-load traffic scenarios [22].

#### 1.2.2. Asynchronous Network Protocols

Sciullo et al. [7] designed an emergency rescue service multi-hop protocol called LOCATE, which uses a personal Internet of Things (IoT) device to provide long range (LoRa) communication to victims where no cellular connectivity is available, with messages re-broadcasted by other peers until reaching a rescue worker who can handle the emergency. Collisions are avoided by assuming a delay-tolerant network in remote environments. Sartori et al. [8] propose a novel approach to creating paths in a multi-hop network by using a newly designed Medium Access Control (MAC) protocol called RLMAC. RLMAC is specifically designed to select the optimal LoRa spreading factor (SF) for each available neighbor, enabling nodes to select the path with the lowest time on-air and reducing overall energy consumption. The authors limit their work however to an asynchronous route discovery, without integrating communication possibilities.

In [9], the authors improve coverage of long-range wide-area network (LoRaWAN) by adding an extra LoRa device in between links, which effectively adds a “hop” between devices. By transferring the authentication by personalisation (ABP) credentials to the hopping nodes, communication remains compatible with the LoRaWAN specification. By adding some extra hops in a LoRaWAN network, thepacket delivery ratio (PDR) and time-on-air (ToA) are improved. Prade et al. [10] mitigate the challenges of collisions and packet loss in asynchronous multi-hop networks by implementing their “Multi-LoRa” architecture: effectively allowing concurrent transmissions and receptions on multiple LoRa frequency channels. Another strategy, to allow for concurrent transmissions, is the use of the property of the spreading technique used in LoRa, i.e., messages sent with a different SF can be distinguished from each other with negligible interference. This is adopted in [12], where each branch of the multi-hop network will transmit using a different LoRa SF, allowing for parallel transmissions between branches.

Finally, [23,24] present a survey of various multi-hop network protocols and architectures for LoRa networks, highlighting their strengths and limitations.

Yet, no open-source asynchronous multi-hop network focuses on ultra low-power operation by exploiting the low data throughput of a remote wireless sensor network (WSN). The work presented in this article, aims to fill that gap.

### 1.3. Requirements of the Intended Applications

**Autonomous and remote network**—The proposed multi-hop network protocol has to be designed with a “*deploy-and-forget*” strategy in mind, particularly for nodes deployed in remote or hard-to-reach locations, such as remote forests, deserts, or offshore platforms, where regular maintenance is difficult and costly. Once a node is properly configured and placed, it should operate autonomously without the need for regular maintenance or monitoring. This can be achieved through features such as low energy consumption, mitigating the need for battery replacements, and self-configuration of the network, i.e., route establishment. Furthermore, the multi-hop network needs to support self-healing by rediscovering the possible routes, after e.g., a relaying node becomes inoperable.

**Low energy consumption**—As elaborated above, the protocol has to be tailored to nodes operating with a stringent energy budget. Strategies such as reducing the “awake” time of any Internet of Things (IoT) node, and reducing the number of transmissions by the efficient composition of payloads need to be implemented to ensure an appropriate life span of the network.

**Operation assumptions**—To attain these goals, the following assumptions with regard to the applicable use cases are made:(1)*Sensor measurement data are limited to at most 12 B.* While this payload may seem small, this is plenty for the aforementioned use cases. In environmental use cases, for example, this could constitute the following payload: 4 B temperature value, 2 B humidity value, 4 B specific gas sensing values for estimating the considered air quality. Together, they would fill only 10 B of the payload.Keeping the data format for sensor measurements short is particularly important to allow for efficient data aggregation. When individual sensor measurement data are kept small, several readings can be aggregated by a ‘hopping’ node into a single message, before being forwarded.(2)*Infrequent sensor sampling.* In aforementioned IoT applications, sensor readings only need to take place sporadically. For example, environmental sensors typically sense slow-changing quantities, and thus only need to send sensor value readings infrequently (typically at most every 30 min).(3)*High latency is acceptable.* In logging applications, the data being collected are typically used for long-term monitoring or historical analysis. The data can be used to understand trends or detect patterns over time. In other applications, such as those in smart farming, high latency may be acceptable due to the nature of the measured variables. Smart farming systems are designed to monitor slow-changing environmental parameters, such as temperature, humidity, soil moisture, and light levels. These variables are typically measured over longer periods of time and are not as time-sensitive as other types of data. Action may only need to be taken within a few days based on the collected data.(4)*All data are sent to a central node: the gateway.* The gateway acts as data sink. Moreover, the network is dense enough so that each node is able to reach the gateway.

### 1.4. Contributions and Manuscript Outline

The work presented in this paper extends our previous research introduced in [25]. Herein, we introduced the novel combination of using preamble sampling and payload aggregation as a valid approach for minimizing energy consumption of long range (LoRa) multi-hop networks. In this work, we detail a full stack communication protocol for such an energy saving multi-hop network. We provide an accompanying multi-hop network simulator, which can be used to optimize the energy-savings in the network. Furthermore, we propose a cross platform network implementation, which is evaluated for both energy efficiency and reliability. To ensure reproducibility and encourage adoption across various application fields, all source files are released as open-source.

This paper is organized as follows (see Figure 2). In Section 2, we provide a detailed explanation of the developed network protocol, including an in-depth description of the operation of the multi-hop platform, such as the automatic route discovery method and multi-hop communication approach. We also discuss energy-saving techniques, such as the use of prolonged preamble sampling and adaptation of the aggregation algorithm.

Next, we present an open-source implementation of the multi-hop network for Arduino-powered embedded systems in Section 3. In Section 4, we explain the accompanying open-source network simulator that can evaluate any multi-hop deployment.

Then, we evaluate the proposed network. We assess the effectiveness of the aggregation algorithm (Section 5.1). Next, we deploy and evaluate a test setup of 30 nodes and determine the optimal parameter set for a general environmental monitoring use case (Section 5.2).

In the conclusion, we summarize the key findings and indicate promising extensions for future work that can improve the reliability and longevity of the multi-hop network.

## 2. Multi-Hop Low-Power Network Protocol

We propose a multi-hop network that is specifically designed to meet the scope and requirements defined in Section 1.3, more specifically, to operate Internet of Things (IoT) devices autonomously in remote locations with minimal energy consumption. In this section, we detail the multi-hop network and explain how the presented protocol is able to meet these requirements.

To reduce the energy consumption of the sensor nodes, the proposed low-power multi-hop networking protocol employs prolonged preambles (Section 2.1). The devices periodically wake up to check the channel activity. If a prolonged preamble is detected, they process the upcoming payload. Otherwise, the nodes go back to sleep, ensuring that receivers can maximize their sleep time. To further improve energy efficiency, a dynamic aggregation mechanism is put in place, where relay nodes aggregate incoming messages and transmit one large packet rather than several small packets (Section 2.5). This way, the energy per effective payload byte in messages is reduced. To enable energy-efficient communication throughout the multi-hop network and still provide autonomous operation, a routing path is dynamically determined for each node (Section 2.3 and Section 2.4).

### 2.1. Prolonged Preamble Sampling

To optimize the energy efficiency of low-throughput low-power wide-area networks (LPWANs), an asynchronous approach outperforms synchronised networks for the use cases under consideration. To mitigate collisions or, in other words, improve the number of successfully received packets, the prolonged preamble sampling technique is used. These preambles are used to detect incoming messages and channel activity. The physical layer protocol LoRa already incorporates preambles to detect and synchronise to an incoming message (see Figure 3). However, in the proposed network protocol, the preamble duration in the LoRa message format is prolonged extensively up to 16 s for LoRa spreading factor (SF) 7, as opposed to the default value of 6.2 ms for long range (LoRa) messages transmitted with spreading factor (SF) 7 (250 kHz bandwidth) [26]. This allows us to create an asynchronous multi-hop network without the need for all nodes to be continuously in RX mode. Instead, nodes will rely on channel activity detection (CAD) to receive messages. To ensure activity is detected, receivers wake up at least twice over the time of one preamble transmission to perform channel activity detection (see Figure 4) In order to reduce collisions between packets transmitted by neighboring nodes, the time interval between two consecutive CAD operations is randomized to a certain degree. This ensures that each node begins transmitting at a different moment in time. Otherwise, neighboring nodes that overhear each other’s messages could become synchronized. This synchronization could lead to interference. By introducing randomness in the timing of CAD operations, the network can more effectively avoid collisions.

This method of preamble sampling is increasingly advantageous with respect to energy consumption for wireless sensor networks (WSNs) where transmission only seldom occurs, i.e., low-throughput networks. By prolonging the preamble time, less frequent CAD cycles are required, lowering the cost of idle listening. Yet, the longer transmission time of the preambles will increase the energy spent in TX mode. This will be addressed later in this paragraph. In the evaluation included in Section 5, we will evaluate whether the prolonged time-on-air (ToA) of each message results in a higher probability of collisions between messages. The imposed duty cycle limits in the applicable bands are regarded as non-critical, as this work focuses on highly infrequent sensor sampling.

The trade-off between energy consumption and preamble duration is depicted in Figure 5 and is highly dependent on the transmit interval. This trade-off is explored by calculating the lifespan of a relaying node for varying preamble duration (calculator tool available online and open-source: dramco.be/tools/lora-multihop, accessed on 11 May 2023). In this calculation, we adopt the assumption that each node forwards every received message exactly once, acting as an intermediate forwarding node in the multi-hop network. Furthermore, if the node has sensor data available, those data are assumed to be appended to that incoming message before it is forwarded. In essence, a single outgoing message is sent for each incoming message. Additionally, we assume a perfect transmission scenario with no packet loss. The lifespan of the node incorporates the total energy consumption: sleep, transmit, channel activity detection (CAD) and receive energy consumption [27]. A clear optimum can be observed for each message interval, e.g., a preamble duration of 3.84 s when sending and receiving messages every two hours. In the presented multi-hop protocol, the Internet of Things (IoT) nodes will perform a CAD at two random times during this preamble duration of 3.84 s. This calculation is further used as an estimation of the most optimal preamble duration for the presented multi-hop network.

The CAD mechanism, implemented in long range (LoRa) modems, is a very power-efficient manner to check the channel for activity. In CAD mode, the modem will perform a fast scan of the band to check for a LoRa preamble. First, the internal phase-locked loop (PLL) will lock on the channel frequency. Secondly, the receiver will record the channel for the time of one symbol length. In this phase, the receiver consumes as much power as in the receiver (RX) phase, however only for the duration of one LoRa symbol:(1)$$T_{\mathrm{sym}}=\frac{2^{\mathrm{SF}}}{\mathrm{BW}}\;,$$
where SF represents the chosen long range (LoRa) spreading factor and BW the applicable LoRa bandwidth. In a network that is configured with SF=7 and BW=500 kHz, the symbol time $T_{\mathrm{sym}}$ would equate to 0.26 ms.

Finally, the digital processor inside the modem will determine the cross-correlation between the channel recording and the ideal preamble waveform. This process takes less time than one symbol, and the power consumption is reduced in this phase. If channel activity is detected, an interrupt is given to the application processor, after which the modem goes back to sleep.

As the channel activity detection (CAD) mechanism will occur rather frequently, it is important to keep its energy consumption, and thus duration, to a strict minimum.

When comparing the lifespan of an Internet of Things (IoT) node that uses prolonged preamble sampling versus continuously listening for incoming packets, one can see tremendous improvements. When comparing the aforementioned values (e.g., sending a message every 2 h on two AA batteries), and considering the energy profile of the hardware (as obtained later in this work, see Section 3), a lifespan of 2.5 years can be achieved. By opening a continuous RX window, the lifespan shrinks to only 2 days.

### 2.2. Frame Structure

In order to enable routing and aggregation in the proposed multi-hop network, a new frame structure (which refers to the contents of the long range (LoRa) payload, as depicted in Figure 3) must be introduced. This new frame structure will be used for all messages in the network (refer to Figure 6). The message header, which is the first part of the frame, is added before the application payload. The message header has a fixed size of 7 bytes (or 8 bytes long when 16-bit addressing is used) and includes the following fields:Msg UID: The first 16 bits are a randomly generated message identifier. This is used by the nodes to detect duplicate messages.Msg type: This byte specifies the type of the message. Currently, three types are supported: (1) ROUTE_DISCOVERY (2) ROUTED_DATA, en (3) DATA_BROADCAST. A detailed description of these different message types will be given further in the text.Hops: The number of hops a message has taken to arrive at a certain node. This field is automatically updated by the nodes when forwarding a message and plays an important role in the route establishment phase.Cumulative LQI: A 16-bit number giving an indication on the quality (hence, link quality indicator (LQI)) and reliability of a certain route in the network (see Section 2.3). In the current implementation, a lower number indicates better quality. When a node forwards a packet it increases the cumulative LQI field with the LQI of the previous hop.Addr: an address field used both in routing and route establishment. For routed data packets, this field contains the destination address. For route discovery packets, this field contains the source address of the transmitting (forwarding) node.

This header consisting of five fields is followed by the payload. The payload is further subdivided into payload blocks, which can be nested into each other as a result of the aggregation algorithm. Each payload block consists of the data sent by each sensor node, accompanied by a small three-field payload header:Src UID: an 8 bit (or 16 bit, when 16-bit addressing is used in the network) unique identifier of the sending node.L1: length of the sensor data generated by this node.L2: length of the data that are generated by other nodes, but are forwarded by this node.

The length of the payload will vary according to the payload bytes required from the general use case (L1) and the node’s position in the network (L2). Thus, the payload data are subdivided in two parts:Data from src: data that are generated by the node considered itself; the length of these data is indicated in field L1.Forwarded data: incoming data that need to be forwarded to other nodes, constituting in a single “hop”. The total length of the forwarded data is recorded in field L2. The forwarded data field will consist of a nested structure of several payload blocks, after several hops. Each payload block is kept distinguishable by a payload header, which incorporates the appropriate lengths.

The above proposed general message structure is optimized to ensure proper operation of the multi-hop network, as will be further validated in the remainder of this work.

### 2.3. Route Establishment

In multi-hop networks, message delivery to a central gateway can be organized in two ways: by flooding a message throughout the network, or establishing routes to the gateway for all nodes. As we consider mostly static networks in this work and optimize for energy efficiency, it is obvious to implement a route establishment protocol in this work (Figure 7). This method should ensure message delivery in the least amount of hops, and opt for a good quality wireless link between hops.

With predetermined intervals, the gateway transmits a ROUTE_DISCOVERY message. Every sensor node that receives this message will re-transmit the package. By doing so, the ROUTE_DISCOVERY propagates through the whole network. By checking the Msg UID, nodes ensure that they transmit a received ROUTE_DISCOVERY packet only once. When a node re-transmits the ROUTE_DISCOVERY message, it adjusts only three fields in the message header: replacing the Src UID with its own UID, incrementing the hop count (Hops field) by one and adjusting the Cumulative LQI field accordingly. Of which the latter is calculated as follows for each route $R_{n}$:(2)$${\mathrm{LQI}}_{R_{n}}=\sum_{l\in R_{n}}{\mathrm{SNR}}_{\max}-{\mathrm{SNR}}_{l},$$
where the set $R_{n}$ contains the links or hops *l* between the gateway and node *n*, ${\mathrm{SNR}}_{\max}$ the maximum signal-to-noise ratio (SNR) by the long range (LoRa) modem and ${\mathrm{SNR}}_{l}$ the SNR over the link *l*. The SNR values are expressed in dB. The maximum SNR (${\mathrm{SNR}}_{\max}$) with the current hardware considered in this paper is 30 dB.

Every sensor can receive the same ROUTE_DISCOVERY message several times, i.e., from all its neighbors, and thus each representing a possible route to the gateway. For each incoming ROUTE_DISCOVERY message, the three most important fields are saved in a circular buffer data structure (i.e., the routing table) of the last eight observed routes: Src UID, Hops, and Cumulative LQI. From this list, the most suited route is chosen for all future ROUTED_DATA messages, until a new ROUTE_DISCOVERY message is received. The route with the lowest Cumulative LQI will be chosen as the most suited route. When the Cumulative LQI is equal for two or more routes, the route which entails the least amount of hops will be chosen. This proposed route establishment mechanism ensures that routed messages are sent to the gateway along the links with the highest accumulated signal-to-noise ratio (SNR), thereby mitigating packet loss due to low signal strength. Furthermore, Equation (Equation 2) ensures that routes are taken with as few hops as possible. Lowering the number of hops and favoring high-signal-to-noise ratio (SNR) links, fewer message forwards and retransmissions are required, improving the energy efficiency of the network.

To clarify the above process, consider the network, shown in Figure 7, which includes one gateway with UID *00* and four sensor nodes with randomized UIDs. The gateway initiates the network by sending a ROUTE_DISCOVERY message. Upon receiving this message, nodes *01* and *02* will update their routing table, and then forward it after a random delay between $\Delta T_{\min}$ and $\Delta T_{\max}$. In the example, node *02* forwards the message first, after updating the Addr, Hops, and Cumulative LQI fields accordingly. Of the nodes that receive this forwarded message, only nodes *03* and *04* will further forward the message, as node *01* already forwarded the original ROUTE_DISCOVERY message. Node *01* will only use the incoming message to update its routing table. Any subsequent messages received by nodes *01*, *02*, *03*, and *04* will not be forwarded as all receiving nodes have already processed the ROUTE_DISCOVERY message. The incoming messages will be used to update their routing tables accordingly. The resulting routing tables are depicted in Figure 8a.

It is clear that routes in the network will remain static between ROUTE_DISCOVERY messages sent by the gateway. The timing of sending these messages from the gateway can be chosen according to the dynamism of the network surroundings: more frequent for changing urban landscapes, and less frequent for more static rural environments.

### 2.4. Message Routing

Routed messages navigate across the multi-hop network to the gateway, according to the route established by the route discovery protocol (described in Section 2.3). When latency-tolerant data are available on a sensor node, they can be sent using ROUTED_DATA messages. Each of these messages will contain the unique identifier of the node that needs to forward this message in the Addr field of the message (from the routing table). When a node receives a ROUTED_DATA message, it will check the Addr field against its own UID. When these match, the Addr field will be updated according to the routing table of the forwarding node and the Hops field will be incremented by one before re-transmitting the incoming message. Other ROUTED_DATA messages will be ignored. Thus, each ROUTED_DATA message will only be forwarded once per hop level. To improve the overall energy efficiency of the multi-hop network, nodes can delay the re-transmission of incoming ROUTED_DATA. Messages that arrive during this extra delay can be merged, hence improving the spent energy per byte.

To clarify, consider the network and example mentioned in Section 2.3 and Figure 7. Node *04* sends data via the multi-hop network to the gateway. The routing table of node *04* states that the most suited route starts with node *03*: the sent ROUTED_DATA message will be addressed to node *03* (i.e., Addr field). Other nodes will ignore the message. After waiting for possible aggregation opportunities (as described in Section 2.5), node *03* will adjust the Addr field accordingly and forward the message to the next “hop”. This mechanism will continue until the gateway (*00*) is reached.

### 2.5. Aggregation to Improve Energy per Byte

One way to reduce the energy cost of the preamble and header transmission is through the use of aggregation techniques [28]. Aggregation allows multiple payloads to be transmitted in a single long range (LoRa) transmission, reducing the relative overhead of the preamble and header bits. This can greatly improve the energy efficiency of the system, especially for applications with a high overhead-to-payload ratio. This is the case in this work, because of the usage of the prolonged preamble at the beginning of each message.

The effect of payload aggregation on the energy consumption by transmission per transmitted sensor reading is displayed in Figure 9. It is clear that energy consumption per sensor reading, for the assumed data length per reading, can be reduced by a factor of seven by fully making use of the available LoRa payload size of 255 B.

Sensor reading aggregation can be achieved locally by aggregating readings from a single node and transmitting them only when necessary. However, further opportunities with regard to aggregation emerge when observing the workings of the multi-hop network. A relaying node will receive sensor readings from multiple sensor nodes and thus is able to aggregate incoming sensor readings as well. In the proposed network protocol, this is implemented with a dynamic aggregation timer: after an intent to transmit, a node will wait for any incoming messages that can be appended to the message currently in the queue to be transmitted.

The aggregation timer begins after the occurrence of a transmission intent (denoted by $t_{n}$, with *n* being the *n*th transmit cycle), either due to sensing or receiving a ROUTED_DATA message, and expires after the specified time $T_{a,n}$ known as the aggregation timer delay. During this time, incoming messages are accumulated and combined into a single message buffer by appending each message’s payload. The aggregated message is then transmitted when the timer expires or the buffer becomes full. Despite its energy-saving benefits, aggregation can result in significant latency in multi-hop wireless sensor networks. As a result, the duration of the aggregation timer adjusts based on network traffic, increasing by $T_{a,\mathrm{upstep}}$ with each incoming message, up to the limit of $T_{a,\max}$. When no messages are received or the buffer is full, the timer decreases by $T_{a,\mathrm{downstep}}$ for the next aggregation cycle, down to the minimum value of $T_{a,\min}$. Thus, the timer duration is computed as,
(3)$$T_{a,n}=\{\begin{array}{cc} \max\left(\left(T_{a,n-1}-T_{a,\mathrm{downstep}}\right),T_{a,\min}\right), & \mathrm{if}M_{n-1}=0\;\mathrm{or}\;\mathrm{full}\;\mathrm{buffer}\\ \min\left(\left(T_{a,n-1}+M_{n-1}\;T_{a,\mathrm{upstep}}\right),T_{a,\max}\right), & \mathrm{otherwise} \end{array}$$
with $M_{n-1}$ the number of received messages during the previous aggregation period $T_{a,n-1}$.

The timer starts at the intended transmission time $t_{n}$, and ends at $t_{\exp,n}$ (as defined by Equation (Equation 4)). A randomized delay, drawn from a uniform distribution (denoted U) between ${-}^{\Delta T}{/}_{2}$ and ${}^{\Delta T}{/}_{2}$, is also added to introduce an additional random delay for every node before transmission.
(4)$$t_{\exp,n}=t_{n}+T_{a,n}+U\left[-{}^{\Delta T}{/}_{2},\;{}^{\Delta T}{/}_{2}\right]$$

This additional delay mitigates the chance of simultaneous transmissions by nodes that have the same aggregation time, hence avoiding collisions and packet loss.

An example timeline on the use of the dynamic payload aggregation model is depicted in Figure 10. The node starts a new message by sensing a sensor and starts the first aggregation timer $T_{a,n}$ (offset by a randomized delay). During $T_{a,n}$, two ROUTED_DATA messages are received that need to be forwarded by the node. When the aggregation timer expires, all payloads are merged into one message that is transmitted. As this message contains two extra payloads, the aggregation timer value that will be used in the future $T_{a,n+1}$, is extended two times by $T_{a,\mathrm{upstep}}$. Next, the node starts a new aggregation timer once another ROUTED_DATA messages is addressed to it. The timer is, again, offset by a random value. As, in this case, no extra messages are received to append, the next aggregation timer $T_{a,n+2}$ will be reduced by one $T_{a,\mathrm{downstep}}$.

## 3. Open-Source IoT Platform Implementation

To accurately evaluate the performance of the network in practical scenarios, we have implemented the presented multi-hop protocol in a networks stack, developed for Arduino-powered embedded systems. The employed hardware [27] and firmware used are both available and open-source (available at github.com/DRAMCO/LoRaMultiHop, accessed on 11 May 2023).

The network stack implementation is designed to run on the “Dramco Uno” platform [29], which is a modification of the “Arduino Uno”, with an added on-board LoRa modem and extra energy-saving measures. This custom board features the Microchip Atmega328p microcontroller and Semtech SX1276 LoRa transceiver. The selection of these components was based on their low cost, availability, and ease of development. The hardware platform is described in detail in [27]. The use of these components allows the firmware to be run on a variety of other platforms without modification. Arduino-based platforms that use the SX1276 can also be targeted with little or no modification to the firmware.

The stack implementation makes use of various available energy-saving techniques, including deep sleep techniques. A fine-grained energy model of this hardware was recorded and is available in Figure 11b. This model includes low-power sleep consumption (23 μW) and periodic channel activity adtection (CAD) cycles (330 μJ), leading to an average idle power consumption of 2.1 mW (when using spreading factor (SF) 7, a long range (LoRa) bandwidth of 250 kHz, and a preamble duration of 1 s). These measurements are used across this publication to calculate various energy-saving impacts and the lifespan of the Internet of Things (IoT) nodes.

## 4. Multi-Hop Network Simulator

To evaluate and optimize network parameters under various conditions, we developed a discrete-event cross-layer simulator in Python based on LoRaEnergySim [28]. The simulator is available as open-source (available at: github.com/DRAMCO/LoRa-multihop-simulator, acessed on 11 May 2023). It models the behavior of Internet of Things (IoT) nodes (e.g., sensing, energy consumption profile) and implements the proposed protocol. The overall topography of the simulator is depicted in Figure 12. In the framework, a *sensor* node will initiate a *message* (containing the gathered sensor data). The message is sent via the *link* interface to the surrounding nodes. The message will proceed through the network according to the *route* protocol (see Section 2.4), before arriving at the *gateway* node.

### 4.1. Nodes

The nodes in the simulator closely follow the network protocol put forth in this work. A node can be one of two types: *sensor* or *gateway*. A sensor node will actively take part in sensing and forwarding sensor data in the network. The gateway operates as a data sink: all collected sensor readings will be stored here.

The operations of each node are characterized by a set of properties and classes (as illustrated in Figure 12): (1) network-related parameters, (2) energy profile, (3) node location, (4) protocol timers, and (5) discovered route.

1.**Network-related parameters.** The specifications of the network behavior are collected in a single class. These include both long range (LoRa) parameters (e.g., spreading factor (SF), sensitivity) and multi-hop protocol parameters (e.g., preamble size, route buffer size). The resulting changes in network behavior can be simulated by iterating through a range of network property values2.**Energy profile.** To accurately determine the energy consumption of each node, a detailed energy profile was constructed based on the power consumption of the in-house developed firmware of this protocol running on the *Dramco Uno* hardware [27]. The same hardware is used in the experimental evaluation in this work (see Section 3). The energy profile is depicted in Figure 11b. Six states are used in the firmware implementation of the protocol: sleep, channel activity detection (CAD), RX, TX (CAD), TX (data), and sensing. The duration and consumed power in each of these states are accurately measured and included in Figure 11b.3.**Node location.** Each node is put into the network on a specific 3D location. Link properties such as the Euclidean distance between two nodes, especially, are calculated based on these values. Other link parameters (e.g., received signal strength indicator (RSSI) and signal-to-noise ratio (SNR)) are derived from the distance in the Link class.4.**Protocol timers.** The proposed network protocol incorporates multiple timers. All timers are pre-configured based on the list of network-related parameters. Both the sensor and gateway nodes use a *sleep timer*, to define the channel activity detection (CAD) intervals. One additional timer is running on the gateway node, i.e., the *route discovery timer*. This timer indicates when a route discovery message needs to be distributed in the network. For sensor nodes, a total of three extra timers are running: *sense timer*, *collision timer*, and *aggregation timer*. The *sense timer* indicates when the node’s onboard sensors values need to be collected. The *collision timer* mitigates collisions by including a back-off window after a CAD cycle determines channel activity. The *aggregation timer* controls the opening and closing of the aggregation window to gather data from multiple nodes in one message (to optimize energy per byte).5.**Discovered route.** Based on the route discovery protocol, each node determines which should be the destination node, i.e., which node should be the recipient of all the messages sent by the node. This destination node and the associated link quality indicator () is stored in the *Discovered route* class. Whenever a new route discovery message is received, this message is passed to the *Discovered route* class to update the destination node.

### 4.2. Messages

Messages in the simulation framework follow the overall fields of the presented network protocol, as depicted in Figure 6: incorporating both header fields and payload lists. When nodes forward a message, forwarded payloads are copied into a payload list in the simulator (similar to Figure 6). To be able to track each message from node to node, each payload in the simulation framework is extended with a list of the identifiers of each forwarding node. This allows the simulator to identify the exact path and thus the exact number of hops a payload needs to travel before arriving at the gateway. Importantly, this extension has no impact on the network’s operations and serves solely as metadata.

### 4.3. Links

Separate link classes exist for each combination of two nodes, which offers the interface of two nodes to send and receive messages. This class consists of two main components: (1) a propagation model to introduce the experienced path loss (PL), and (2) an signal–to-noise ratio (SNR) model to include noise.

1.**Propagation model.** The simulation framework offers, by default, three channel models, based on measurements obtained in multiple environments: open/coastal, forested, and urban environments from [30] (Note that other PL models can be easily implemented in the framework, and thus the framework is not limited to the here proposed models). They are based on a log-distance PL model with a per-link shadowing factor induced by e.g., blockage. The log-normal path loss PL at a distance *d* in 2D space can be expressed as:
(5)$$\mathrm{PL}\left(d\right)=\mathrm{PL}\left(d_{0}\right)+10n\log\left(\frac{d}{d_{0}}\right)+X_{\sigma }\;,$$
where $d_{0}$ is the reference distance and thus the minimal distance between transmitter and receiver, *n* is the path loss exponent and $X_{\sigma }$ models the shadowing component. The shadowing component follows a zero-mean Gaussian distributed random variable with standard deviation σ, $X_{\sigma }∼N\left(0,\sigma \right)$. The parameters for the different environments are summarized in Table 2. The considered PL models deviate from conventional PL models proposed by 3GPP and others, as the network topology differs from the assumed topologies, i.e., the nodes are commonly placed on the same height rather than having an elevated height for the gateway as assumed in other work.2.**SNR model.** In the standard operation, the simulator framework takes thermal noise [31] into account to compute the SNR of the signal, i.e.,
(6)$${\mathrm{SNR}}_{\mathrm{dB}}=\mathrm{TP}-\mathrm{PL}-N\;,$$
with TP the transmit power (in dBm) and *N* the additive white Gaussian noise (AWGN) power (in dB). The noise power is computed as $N=k_{B}T\Delta f$ with $k_{B}$ the Boltzmann constant, *T* the temperature and Δf the signal bandwidth. Given that we use the long range (LoRa) PHY with a bandwidth of 500 kHz and assume a temperature of 25 °C Equation (Equation 6) becomes
(7)$${\mathrm{SNR}}_{\mathrm{dB}}=\mathrm{TP}-\mathrm{PL}+116.87\;.$$

In future versions, more complex link models can be included. For example, the LoRa simulator presented by Al Homssi et al. in [32] includes an accurate interference algorithm and channel emulator which employs ray-tracing to accurately depict the effect of multi-path fading.

### 4.4. Output

The simulation framework can provide valuable insights into the workings of the presented multi-hop protocol. For any set of network parameters, the following resulting metrics can be simulated.

**Packet delivery ratio**: The framework provides both the overall and per-node packet delivery ratio (PDR) w.r.t. a node’s position in the network. This takes both the path loss model (described in Section 4.3) and potential collisions with transmissions from neighboring nodes into account.**Energy models**: All nodes in the network have access to an in-depth energy profile of the used hardware. As a result, energy-related metrics can be simulated: energy consumed by one node (w.r.t. the position in the network), energy load across the network, and the energy consumed per packet (and thus per byte or per payload).**Latency**: The time between the intent of sending a sensor value and the time of receiving the data on the gateway (i.e., latency) is recorded on a per-payload basis.**Aggregation ratio**: The aggregation ratio $\alpha_{\mathrm{aggregation}}$ is defined as the ratio of the messages sent by one node that included payloads from other messages ($n_{\mathrm{aggregated}}$) to the total number of sent messages ($n_{\mathrm{total}})$:
(8)$$\alpha_{\mathrm{aggregation}}=\frac{n_{\mathrm{aggregated}}}{n_{\mathrm{total}}}\;.$$

A lower value of $\alpha_{\mathrm{aggregation}}$ implies that fewer messages are carrying forwarded data.

The framework enables simulation of the impact of various network parameters (including node positions, network density, and payload size) on the metrics mentioned. This is achieved through Monte Carlo simulations, where a set of randomly selected network values within a given distribution is repeatedly simulated to observe the effects on the metrics.

## 5. Evaluation

The proposed network protocol is analyzed by conducting two experiments. First, we evaluate the effects on the energy efficiency of the proposed aggregation methods, in a real-life demonstration network. In our second experiment, by using the simulation framework, we are able to establish an ideal set of network settings (w.r.t. energy efficiency) for any use case. These parameters are validated experimentally in a local campus-wide test.

### 5.1. Experiment 1: Impact of Performing Aggregation in a Multi-Hop Network

The aggregation feature that has been implemented in the proposed multi-hop network is expected to significantly reduce energy consumption and increase the resulting data latency. In order to better understand this impact, we study the effect of aggregation at multiple points in the network on these two parameters experimentally. To ensure consistency, we establish a fixed network topology and configuration: an 18-node Internet of Things (IoT) network with a randomly generated topology, as depicted in Figure 13, overriding the automatic route discovery mechanism. Other relevant network parameters are listed in Table 3.

#### 5.1.1. Reducing Energy Consumption by Aggregation

The aggregation protocol is implemented in the deployed multi-hop network to improve the energy consumption per byte. Herein, we evaluate the aggregation ratio and resulting energy savings experimentally. The results for relaying nodes are summarized in Table 4.

We use two metrics to compare the energy consumption of the network with and without aggregation. $E_{B,\mathrm{no}\;\mathrm{aggregation}}$ represents the average energy consumption per payload byte when aggregation is disabled in the network, while $E_{B,\mathrm{aggregation}}$ indicates the energy consumption per payload byte when aggregation is active in the network.

In our study, nodes *06* and *16* exhibited relatively low aggregation ratios, with approximately half of their transmitted messages containing forwarded payloads. This is expected, as these nodes have only one child node sending data to them, which limits the potential for aggregation. Nodes *02* and *04* demonstrate a high aggregation ratio, which leads to substantial energy savings when aggregation is enabled for the network. Our results confirm that energy savings are most pronounced for nodes with a high number of children. Specifically, in our study, the sensor node that forwards data of four children is able to reduce its energy consumption by up to 61% per transmitted message by leveraging data aggregation.

#### 5.1.2. Reducing Latency by Dynamic Aggregation

Increased latency is an inherent drawback of using aggregation in a multi-hop network. To combat adding unnecessary delays to nodes that do not forward data due to their position in the network, a dynamic timer mechanism is put in place (see Section 2.5). To validate this mechanism, several nodes in the fixed network depicted in Figure 13 are logged to keep track of timer value variations. The evolution of the aggregation time $T_{a}$ is depicted in Figure 14. Both axes, in this figure, have been normalized to the measurement delay $T_{\mathrm{measure}}$, as the aggregation timer $T_{a}$ varies depending on the availability of data (and thus the measurement interval $T_{\mathrm{measure}}$).

It is clear that all nodes eventually opt for either the maximal aggregation delay $T_{a,\max}$ (maximizing aggregation ratio $\alpha_{\mathrm{aggregation}}$) or the minimal $T_{a,\min}$. Nodes that receive a lot of incoming data during the aggregation window, will quickly saturate on $T_{a,\max}$. This stems from the fact that the protocol has been designed with energy efficiency in mind. Nodes that forward data need to increase their chance of accumulating data. This is what happens, for example, with node *04* in Figure 13. This node will directly forward data from four other child nodes: $T_{a,\max}$ will be reached after only one measurement interval $T_{\mathrm{measure}}$. Other nodes which, for example, only forward data from one child node (such as node *06* or *16* in Figure 13) will need more time to saturate to the maximal aggregation timer value $T_{a,\max}$. Nodes that do not forward data, i.e., leaf nodes, will never aggregate data. As such, their aggregation timer will evolve to the minimal aggregation timer value $T_{a,\min}$ and minimize latency from the leaf node to the gateway.

To quickly maximize the aggregation potential, the $T_{a}$ is increased with a step size of $T_{a,upstep}$. To keep it maximized for nodes with a lower aggregation ratio, the decrease of $T_{a}$, denoted $T_{a,downstep}$ is taken smaller as $T_{a,upstep}$ (see Table 3 for the values used). This can also be seen in Figure 14. However, when a node stops forwarding data due to a change in the routing, e.g., because of a change in the environment or the network itself, its $T_{a}$ would evolve to $T_{a,min}$ over time.

Our results (see Section 5.2.1) show that this mechanism indeed optimizes the energy consumption of the aggregating nodes.

### 5.2. Experiment 2: University Campus Deployment

The proposed multi-hop network is evaluated based on a series of experiments in real-world settings. We deployed the experimental platform across the university campus, taking measurements and collecting data to assess the network’s performance under various scenarios. However, to identify the most optimal network settings prior to deployment, we first conducted simulations using the network simulator presented in this study. By running simulations with different configurations and traffic loads, we were able to evaluate the network’s behavior and identify the best settings that would maximize its performance while minimizing energy consumption and other resources. This is discussed in Section 5.2.1.

The deployed network is depicted in Figure 15. In total, 30 nodes and one gateway are deployed across campus, both indoors and outdoors, comprising a total area of 200 m by 170 m with some indoor sensor nodes placed at different floors. The long range (LoRa) parameters are set to the settings providing the lowest range, i.e., output power of 0 dBm and spreading factor (SF) 7. In this way, we intentionally emulate a more harsh propagation environment, similar to the conditions present in a more large-spread deployment. The range is further reduced by placing most of the nodes indoors. This is to stress-test the proposed protocol and illustrate its applicability to the mentioned target applications.

#### 5.2.1. Determining Optimal Configuration via the Simulator

To explore the impact of various parameters on the workings of the presented long range (LoRa) multi-hop network, we simulate the network depicted in Figure 15 by using the simulation framework described in Section 4. All links are simulated using the urban path loss model (as noted in Table 2). Criteria on the performance of any parameter combination are the resulting packet delivery ratio (PDR), packet loss ratio (PLR) due to collisions, energy consumption, and latency. In particular, we simulate the effects on the performance of the network of the following parameters: (1) network throughput, (2) aggregation time, (3) randomization parameters, and (4) payload size.

1.**Network throughput.** In the current implementation of the presented multi-hop wireless sensor network (WSN), the size of generated sensor data is fixed for all nodes in the network. As a result, network throughput is highly dependent on the rate at which data are generated by the Internet of Things (IoT) nodes. This rate is determined by the measurement timer $T_{\mathrm{measure}}$ used to capture sensor data. Lower values of $T_{\mathrm{measure}}$ will lead to higher throughput, possibly saturating the network due to a large number of collisions caused by a longer time-on-air (ToA) per packet, as dictated by the employed prolonged sampling mechanism (see Section 2.1). This is indeed the case. The results in Figure 16 show that, for values of $T_{\mathrm{measure}}$ below 6.5 min, the packet delivery ratio (PDR) decreases significantly as $T_{\mathrm{measure}}$ decreases further. When $T_{\mathrm{measure}}$ is set to 3 min, for example, nodes with three child nodes experience a drop in PDR to only 52%. It is worth noting that this simulation takes the energy optimization w.r.t. the preamble duration into account (as discussed in Section 2.1). As such, preamble duration is adjusted throughout the graph to the optimal value according to $T_{\mathrm{measure}}$. Evidently, energy consumption increases with an increased network throughput as more and longer messages will be sent in the network.2.**Aggregation time.** As discussed in Section 5.1, increasing the aggregation ratio $\alpha_{aggregation}$ can reduce energy consumption for all forwarding nodes in a multi-hop network. It is well established that extending the time that nodes wait for incoming messages to aggregate (i.e., increasing the aggregation timer $T_{a}$) can benefit aggregation and energy efficiency. Figure 17 and Figure 18 show the relationship between $T_{a}$ and $\alpha_{aggregation}$ and energy efficiency, respectively, for nodes throughout the network.Figure 17 demonstrates that $\alpha_{aggregation}$ increases with an increase in $T_{a}$. However, nodes with a larger number of child nodes (two or more, in this example) experience a more gradual increase in $\alpha_{aggregation}$ because the likelihood of data being forwarded from one of their child nodes increases with the number of child nodes. The aggregation ratio stabilizes for all nodes at around 90% when $T_{a}/T_{\mathrm{measure}}=0.5$.The energy per byte spent on transmitting data ($E_{\mathrm{TX},B}$) is closely linked to the aggregation ratio $\alpha_{\mathrm{aggregation}}$ as more aggregation will cause $E_{\mathrm{TX},B}$ to decrease. Yet, $E_{\mathrm{TX},B}$ inherently also takes the size and amount of the aggregated data into account. As shown in Figure 18, as more message aggregation takes place, less energy per byte is required, leading to overall energy savings in the multi-hop network. It is worth noting that the potential energy savings increase when a node has more child nodes, as more data can be aggregated. $E_{\mathrm{TX},B}$ stabilizes for all forwarding nodes at $T_{a}/T_{\mathrm{measure}}=0.5$. There, a saving of 69% in energy consumption is achieved. By maximizing $T_{a}$, nodes with more than two child nodes can save up to 75% of transmission energy, yet at the expense of a disproportionate extra latency.However, increasing $T_{a}$ also results in an increasingly higher latency, which is disproportionate to the energy savings already obtained at $T_{a}/T_{\mathrm{measure}}=0.5$ (69%).It is worth noting that simulations also demonstrate how leaf nodes can save additional energy. By aggregating messages, fewer messages are transmitted, which minimizes overhearing by leaf nodes. As a result, less energy is consumed during the receive process, leading to a possible energy reduction of up to 11%.

**Figure 17 sensors-23-04994-f017:**
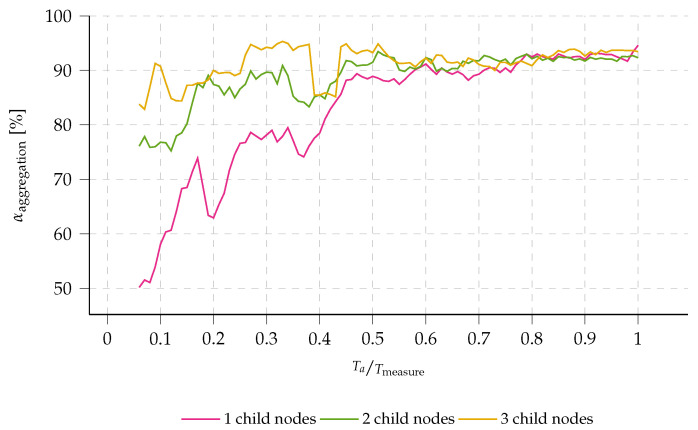
Simulated aggregation ratio $\alpha_{\mathrm{aggregation}}$ across the multi-hop network w.r.t. the ratio of the aggregation timer value $T_{a}$ to $T_{\mathrm{measure}}$. It is clear that the aggregation ratio $\alpha_{\mathrm{aggregation}}$ increases with an increased $T_{a}$, up until $T_{a}/T_{\mathrm{measure}}=0.5$, after which the aggregation ratio stabilizes around 90%.

**Figure 18 sensors-23-04994-f018:**
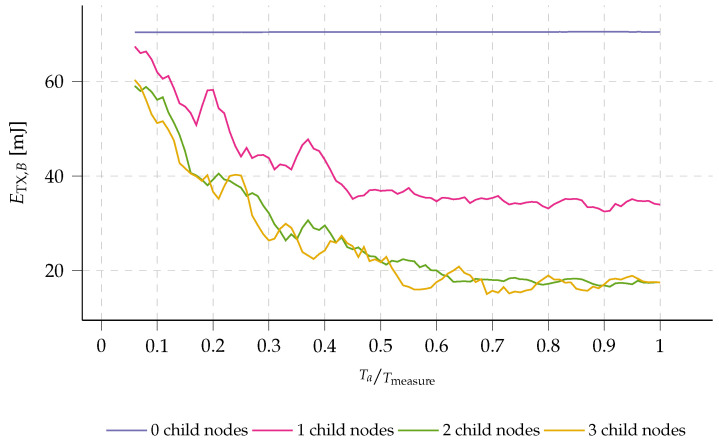
Simulated energy per byte spent by transmitting data $E_{\mathrm{TX},B}$ across the multi-hop network w.r.t. the ratio of the aggregation timer value $T_{a}$ to $T_{\mathrm{measure}}$. It is clear that, for forwarding nodes, when $T_{a}$ increases, the energy $E_{\mathrm{TX},B}$ decreases significantly until $T_{a}/T_{\mathrm{measure}}=0.5$, after which it stabilizes at a minimum. This is caused by the increase and stabilization of the aggregation ratio $\alpha_{\mathrm{aggregation}}$ (as observed in Figure 17). By maximizing $T_{a}$, up to 65% transmission energy can be saved for nodes with more than 2 child nodes.

3.**Randomization parameters** To prevent messages from colliding with those transmitted by nearby nodes, some network timers use randomized values for each cycle. For example, the time between two consecutive channel activity detection (CAD) cycles and the aggregation timer includes a randomization margin. Simulations have shown that the randomization of these timers has minimal impact on network performance in terms of latency or energy consumption. It is key to introduce randomization. Simulations show that the degree of randomization has a very low impact on the packet delivery ratio (PDR). The results for these simulations are published in the open source repository of the multi-hop simulator: github.com/DRAMCO/LoRa-multihop-simulator, last accessed on 11 May 2023.4.**Sensor data size.** Increasing the data size in the presented multi-hop network will result in higher energy consumption for all nodes since this increases the airtime required to transmit the data. In addition, simulations have shown that when the data size is increased to, for example, 64 bytes, the aggregation ratio $\alpha_{\mathrm{aggregation}}$ can decrease by up to 9%. This is because the aggregation buffers fill up faster, reducing the amount of aggregation that can take place and leading to a further slight increase in energy consumption. Despite the prolonged airtime, the network’s packet delivery ratio (PDR) is not significantly affected by increasing the data size, according to simulations. The results for these simulations are published in the open source repository of the multi-hop simulator: github.com/DRAMCO/LoRa-multihop-simulator, last accessed on 11 May 2023.

The simulations presented above provide valuable insights into the configuration of the deployed multi-hop network. Based on these insights, we determine the optimal network parameters for deployment, as summarized in Table 5. For general Internet of Things (IoT) monitoring use cases, where fast update rates are not required, we have established a measurement interval of $T_{\mathrm{measure}}=30$ min to conserve energy. To balance energy efficiency and latency, the aggregation timer ($T_{a,\max}$) has been set to half of the measurement interval, i.e., 15 min. To minimize network latency, the minimum aggregation timer ($T_{a,\min}$) has been set to 0 min. The size of sensor measurements has been fixed at 12 B, as previously set in Section 1.3.

#### 5.2.2. Real-World Multi-Hop Performance

In order to evaluate the performance of the multi-hop platform presented in this study, we deployed 30 sensor nodes and one gateway across the university campus (see Figure 15). The deployed sensors and gateway are based on the “Dramco Uno” platform as described in Section 3. Each sensor node was configured to transmit an incrementing counter, enabling us to assess criteria such as the packet delivery ratio (PDR) for any given node. The data received at the gateway node (node *00*) were logged for subsequent analysis. The network was configured as listed in Table 5. The experiments were conducted in a 48 h window.

The first step taken by the network is self-configuration, i.e., determining the optimal route for message transmission. This process creates a tree-structured network as shown in Figure 19.

The majority of the network’s nodes are situated at a distance of zero or one hop from the gateway. Three nodes are located at two hops from the gateway, and another three are at three hops.

The gateway sends out a route discovery message every 6 h (as defined by $T_{\mathrm{route}}$ in Table 4). Therefore, the routes for each node may change every 6 h due to varying signal-to-noise ratio (SNR) values across links (see Section 2.3). In Figure 19, the path taken by the majority of messages is represented in black, while alternative paths, taken by the minority of messages, are colored grey.

It should be noted that for some links, the trade-off between adding multiple hops with a good SNR and adding (multiple) extra hops to the communication is not straightforward for some nodes: these switch between two options, highly based on ${\mathrm{SNR}}_{l}$. For example, node *18* is able to communicate directly with the gateway or via nodes *17* and *23*, which adds two extra hops to the path. Interestingly, *18* sends out 68% of messages via the extra hops, and only 32% of messages directly to the gateway *00*, even though the packet delivery ratio (PDR) for both paths is the same (88%).

During the 48 h logging period, the network’s PDR was monitored for each node. The cumulative distribution function (CDF) for the PDR is shown in Figure 20. The results indicate that 90% of the nodes achieved a PDR of at least 70%. However, when analyzing the PDR in relation to the node’s distance from the gateway, in terms of number of hops, it is clear that nodes located farther away from the gateway suffer from a lower PDR. The decrease in PDR observed in the presented multi-hop network with an aggregation scheme is due to its inherent limitations. In any multi-hop network, the average PDR of a message is the product of the average PDR values of all traversed links:(9)$$\mathrm{PDR}=\prod_{l}^{L}{\mathrm{PDR}}_{l},$$
with ${\mathrm{PDR}}_{l}$ the PDR at link *l*. Additionally, since our aggregation scheme combines multiple sensor data measurements into a single message, a single lost message due to collision or other reasons can result in substantial data loss.

The effectiveness of the aggregation mechanism is also evaluated in the experimental test setup. The aggregation ratio $\alpha_{\mathrm{aggregation}}$ is determined for each node. The results are depicted in Figure 21: the cumulative distribution function (CDF) of the aggregation ratio $\alpha_{\mathrm{aggregation}}$, with respect to the number of forwarding child nodes. The results show the effectiveness of the aggregation mechanism with the dynamic aggregation timer (as discussed in Section 2.5). Nodes with no child nodes, evidently do not forward any data. The chance that messages are received within the aggregation time, increases with the amount of forwarding child nodes. Yet, as the aggregation timer maximum $T_{a,\max}$ is set at $T_{\mathrm{measure}}/2$ (as determined in Section 5.2.1), some nodes will still transmit messages that do not contain aggregated data. This is especially the case for nodes with one or two child nodes. Nodes that forward data from three child nodes, have an aggregation ratio of at least 90%, effectively improving the energy per byte, as evaluated in Section 5.1 and Table 4.

## 6. Conclusions

In this work we presented a low-power and reliable multi-hop long range (LoRa) network. By combining and optimizing prolonged preamble transmission and periodic channel activity detection (CAD), a node’s lifespan is ensured, and is still able to forward messages in the multi-hop network. For example, when nodes transmit data only every 6 h, the lifespan can be increased from up to 2 days to up to 4 years. Furthermore, we proposed a cross-platform and open-source network protocol which further reduces energy consumption by smartly routing and aggregating messages in between hops. A simulator framework to assess the effects of network configurations is presented. This multi-hop protocol is experimentally validated to prove both the possible energy gains and the reliability. By employing the discussed dynamic aggregation mechanism, a further TX energy consumption reduction of up to 61% can be obtained. Communication in the presented network is proved to be reliable: 90% of nodes achieve a packet delivery ratio (PDR) of at least 70%.

Future research could explore other mechanisms to determine the optimal routes between multiple neighbors, for example RL-MAC [33], which employs reinforcement learning to optimize data throughput and energy consumption in wireless sensor networks (WSNs). Furthermore, optimizing network load across multiple nodes is not discussed in the current implementation of the multi-hop protocol. By incorporating both the remaining available energy and the current communication load of an Internet of Things (IoT) node in the route discovery mechanism, a more even distribution of energy consumption across the network can be obtained. In order to improve the reliability of the network, acknowledgments and re-transmissions might be employed, positively affecting the PDR. Finally, a data broadcast mechanism might be implemented in order to quickly relay time-critical messages such as alarms from nodes to the gateway.

## Figures and Tables

**Figure 1 sensors-23-04994-f001:**
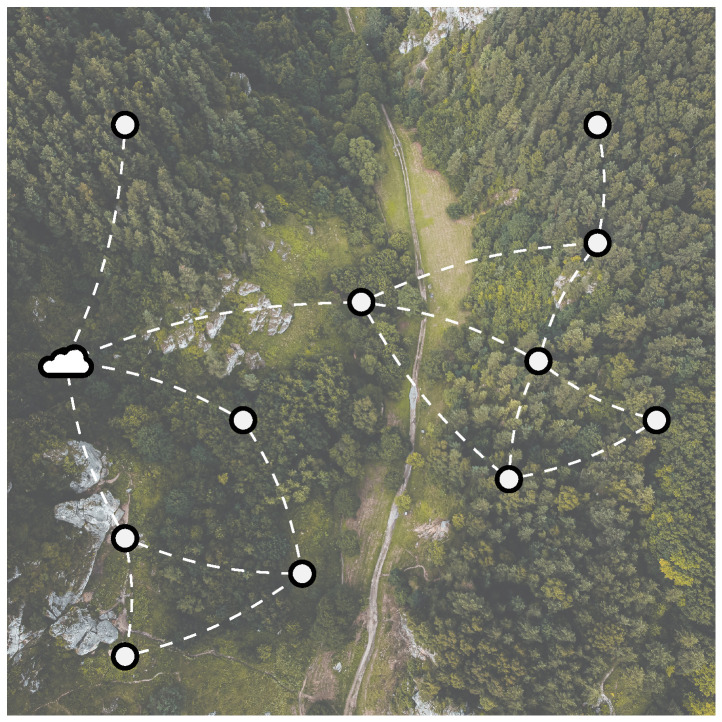
A multi-hop network deployment in a remote environment, designed for environmental tracking purposes in forests or nature reserves.

**Figure 2 sensors-23-04994-f002:**
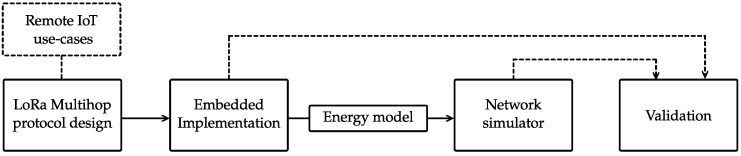
Overview of the conducted research: from use-case applications to the validation of the presented multi-hop network. The dotted lines represent inputs to the research block in question.

**Figure 3 sensors-23-04994-f003:**
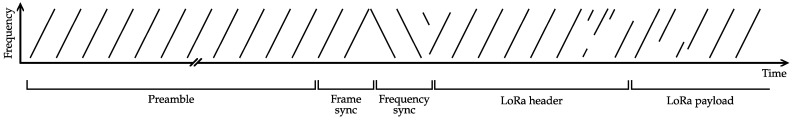
Long range (LoRa) chirp spread spectrum (CSS) illustration with annotated message fields. The cyclic redundancy check (CRC) field after the long range (LoRa) payload has been omitted in this figure.

**Figure 4 sensors-23-04994-f004:**
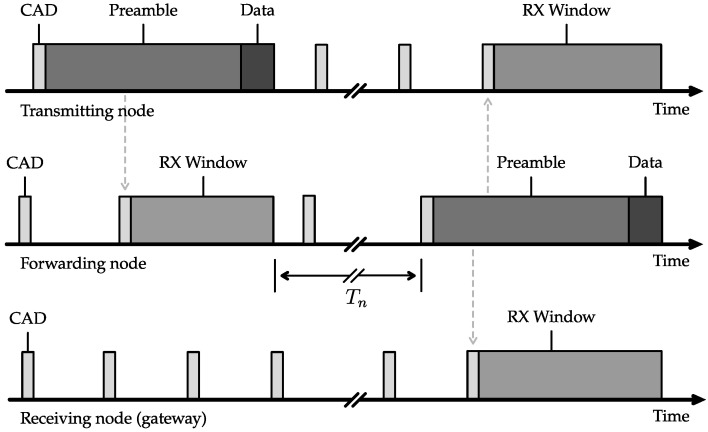
Illustration of how the prolonged preamble sampling technique can be employed in a multi-hop network.

**Figure 5 sensors-23-04994-f005:**
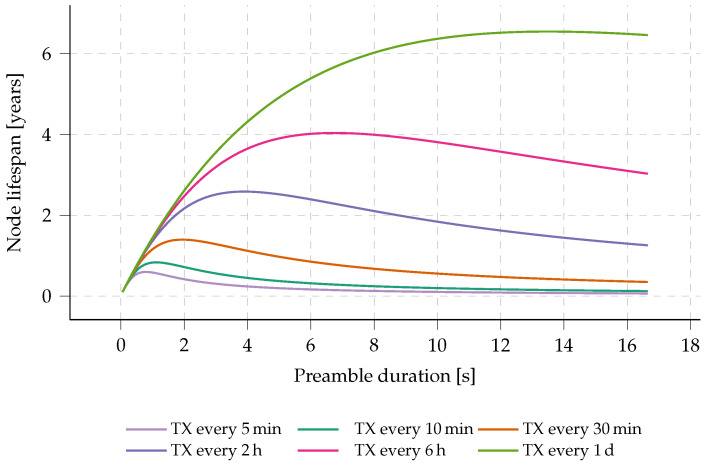
Exploration of the lifespan of a multi-hop sensor node versus the set preamble duration. Calculations based on equal amount of RX and TX, SF7, 500 kHz bandwidth and, 30 B payload. The chosen battery size is equal to the capacity of two common AA batteries: 2500 mAh. Note that this assessment does not incorporate the self-discharge rate of the batteries.

**Figure 6 sensors-23-04994-f006:**
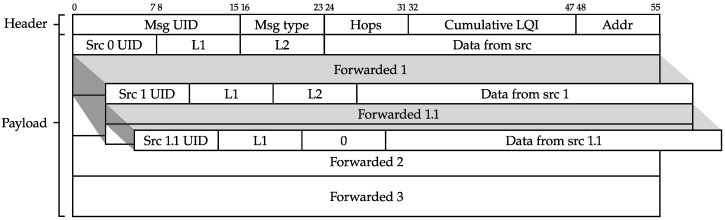
General message structure. The header with five fields (7 B) is followed by the payload. Depending on the type of packet and a node’s position in the network, the payload can have a nested structure.

**Figure 7 sensors-23-04994-f007:**
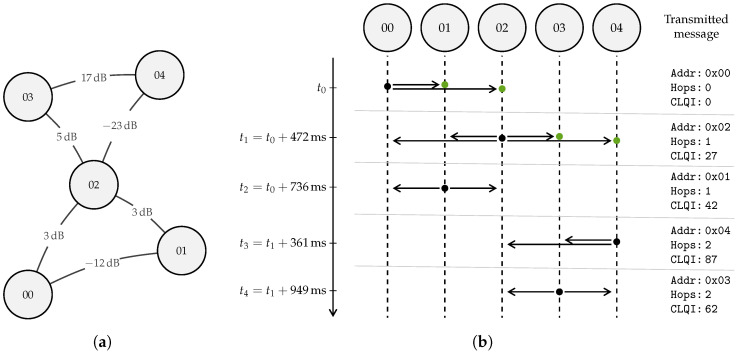
Example of the route discovery mechanism. (**a**) Example network topology: the SNR value of each link is depicted between Internet of Things (IoT) nodes. (**b**) Timeline of the route discovery messages propagation through the network. Each forwarded message (i.e., hop) occurs at a random time ($U\left[\Delta T_{\min},\Delta T_{\max}\right]$) after message reception. Only relevant message fields of the transmitted messages are shown.

**Figure 8 sensors-23-04994-f008:**
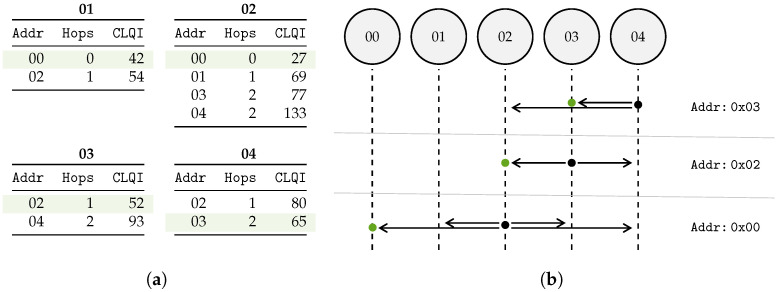
Example of how data are routed to the gateway, using the network topology from Figure 7a. (**a**) The resulting routing tables of each sensor node in the network, the best route is selected by the lowest Cumulative LQI field, as highlighted. (**b**) Routing of a message sent by sensor node *04*. Only the node with the appropriate UID (i.e., Addr) will forward the message until the gateway *00* is reached.

**Figure 9 sensors-23-04994-f009:**
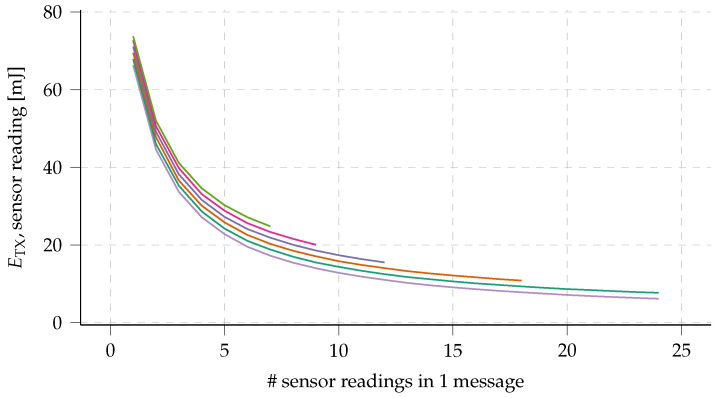
Effect of aggregating multiple sensor readings in one message on the energy consumption. Energy consumption is displayed as the energy consumed by transmitting the aggregated message, divided by the number of sensor readings in said message. Calculated for multiple sensor reading sizes, limited by the maximum long range (LoRa) message size (255 B), and with a 1 s preamble length.

**Figure 10 sensors-23-04994-f010:**
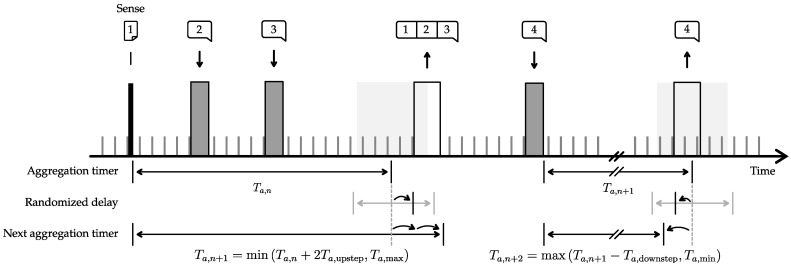
Payload aggregation model as implemented in the presented multi-hop network: minimizing energy consumption by utilizing payload aggregation and minimizing the resulting latency by varying the aggregation timer based on the network throughput. The expected transmission time of a message, denoted as $t_{\exp,n}$, is determined by a combination of $T_{a,n}$, the aggregation timer as defined in Equation (Equation 3), and a randomized delay introduced by $U\left[-{}^{\Delta T}{/}_{2},\;{}^{\Delta T}{/}_{2}\right]$ (Equation (Equation 4)).

**Figure 11 sensors-23-04994-f011:**
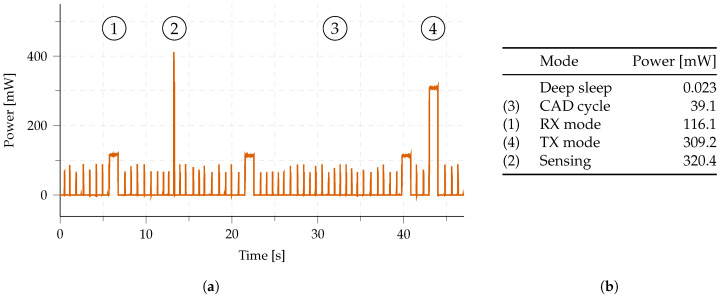
Measured power profile of the implemented long range (LoRa) multi-hop network (**a**), operating at spreading factor (SF) 7. These stages can be observed, alternated with deep sleep (**b**): (1) RX mode, (2) sensing mode, (3) repetitive channel activity detection (CAD) cycles, and (4) TX mode.

**Figure 12 sensors-23-04994-f012:**
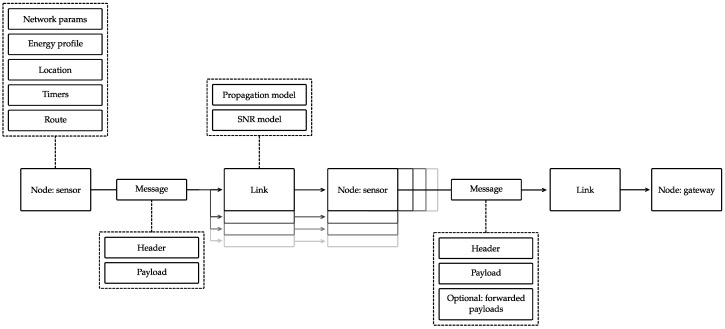
Overview of the network simulation framework structure, used to evaluate the proposed protocol.

**Figure 13 sensors-23-04994-f013:**
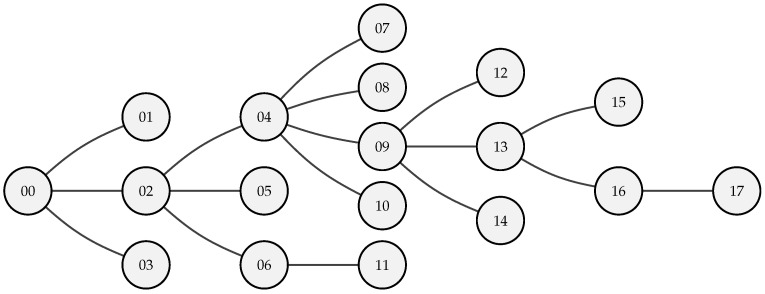
Fixed network topology to study the impact of aggregation across the multi-hop network.

**Figure 14 sensors-23-04994-f014:**
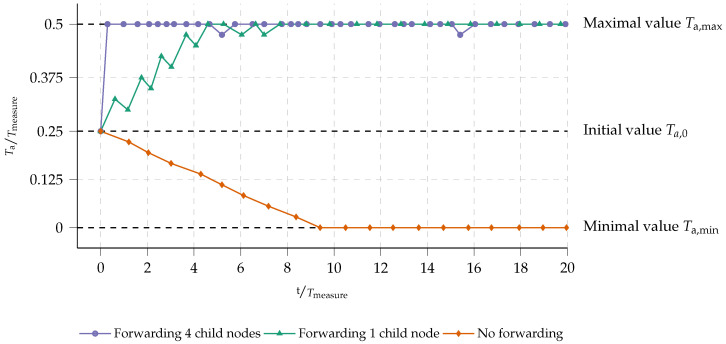
Experimentally validated convergence of the aggregation timer, normalized to the measurement delay $T_{\mathrm{measure}}$.

**Figure 15 sensors-23-04994-f015:**
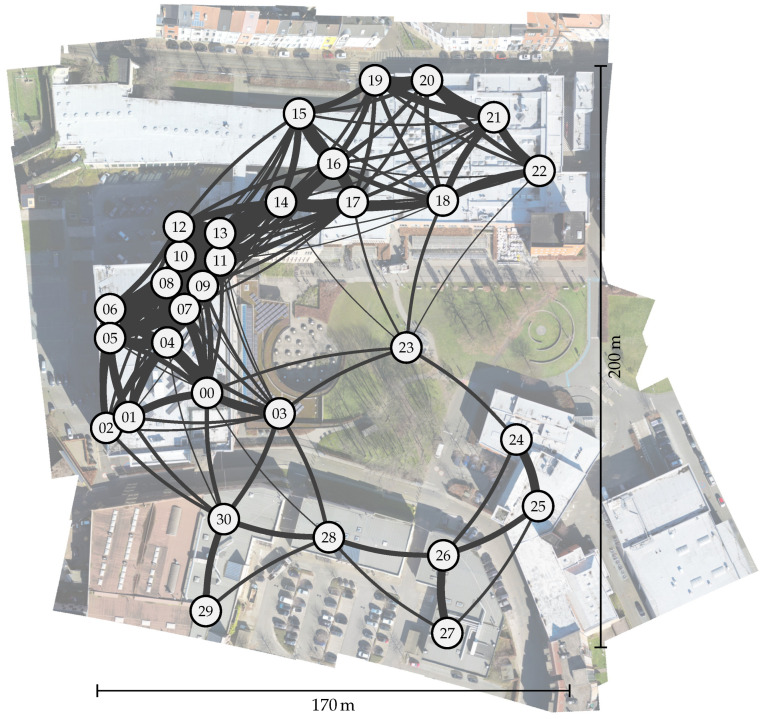
Overview of the placement of the Internet of Things (IoT) nodes in the university campus deployment.

**Figure 16 sensors-23-04994-f016:**
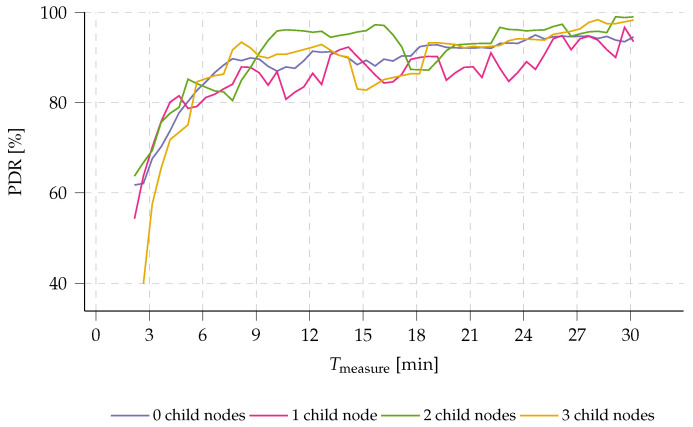
Simulated packet delivery ratio (PDR) across the multi-hop network plotted w.r.t. the measurement interval $T_{\mathrm{measure}}$. Lower $T_{\mathrm{measure}}$ implicates a higher throughput: it is clear that the network throughput degrades when lowering $T_{\mathrm{measure}}$ further than 6.5 min. Any faster messaging results in a drop of PDR across the network.

**Figure 19 sensors-23-04994-f019:**
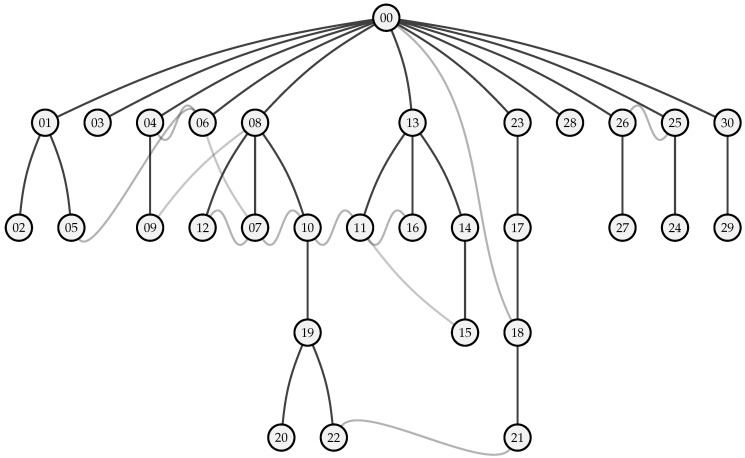
Tree diagram depicting the multi-hop routes established by the route discovery algorithm, based on experimental data gathered from the messages that have arrived at the gateway node (node *00*). The diagram shows the routes that the majority of the messages have traveled, with alternative routes indicated in grey that were taken by a minority of messages. In this test, the route discovery mechanism is updated every 6 h during the 48 h test period. Node positions correspond to those reported in Figure 15.

**Figure 20 sensors-23-04994-f020:**
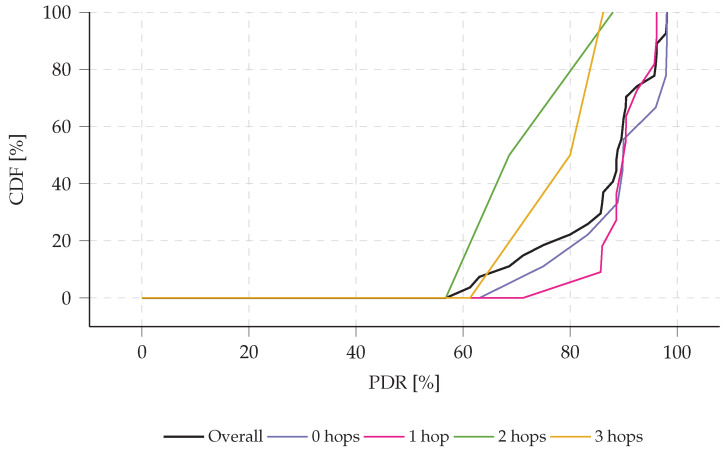
Cumulative distribution function (CDF) of the experimentally determined packet delivery ratio (PDR) with respect to the node’s position in the network (as shown in Figure 19). The results indicate that nodes that require two or three hops before reaching the gateway tend to have a lower packet delivery ratio (PDR) compared to other nodes. When such messages are aggregated with other sensor data, the loss of a single message may result in a substantial drop in PDR.

**Figure 21 sensors-23-04994-f021:**
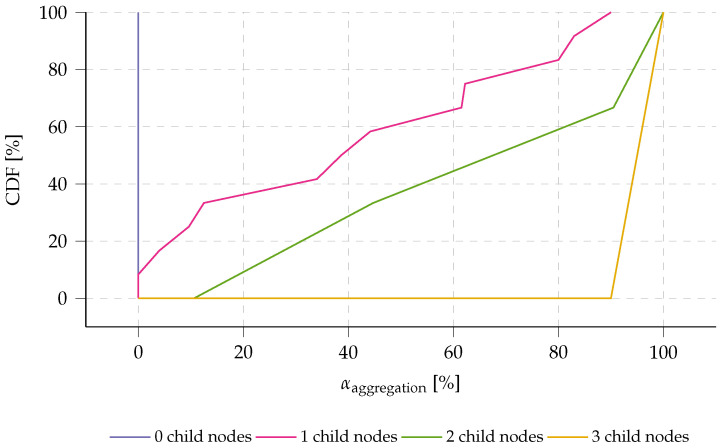
The Cumulative distribution function (CDF) of the experimentally validated aggregation ratio ($\alpha_{\mathrm{aggregation}}$) is shown with respect to the number of forwarding children and their positions in the network (see Figure 19). The results demonstrate the effectiveness of the aggregation mechanism, where nodes with three child nodes show a 90% rate of forwarded data in their messages, resulting in significant improvements in energy per byte.

**Table 2 sensors-23-04994-t002:** Path loss (PL) parameters for each considered environment (based on [30]).

Environment	$d_{0}\;\left[m\right]$	$\mathrm{PL}\left(d_{0}\right)\;\left[\mathrm{dB}\right]$	*n*	$\sigma \;\left[\mathrm{dB}\right]$
Open/coastal	1	43.96	3.62	27.51
Forested	1	95.52	2.03	6.87
Urban	1	74.85	2.75	11.25

**Table 3 sensors-23-04994-t003:** Network parameters used to study the impact of aggregation across the multi-hop network.

Parameter	Symbol	Value
Measure interval	$T_{\mathrm{measure}}$	10 min
Minimal aggregation timer	$T_{a,\min}$	0 s
Initial aggregation timer	$T_{a,0}$	2.5 min
Maximal aggregation timer	$T_{a,\max}$	5 min
Aggregation upstep	$T_{a,\mathrm{upstep}}$	1 min
Aggregation downstep	$T_{a,\mathrm{downstep}}$	30 s
TX buffer size		150 B
Payload size		6 B
LoRa spreading factor	SF	7
LoRa bandwidth	BW	500 kHz
Preamble width	$T_{\mathrm{preamble}}$	1 s

**Table 4 sensors-23-04994-t004:** Impact of the proposed aggregation mechanism on the energy consumption of relaying nodes in the network, depicted in Figure 13 (which is using the configuration noted in Table 3).

Node UID	# Children	$\alpha_{\mathrm{aggregation}}$	$E_{\mathrm{TX},B,\mathrm{no}\;\mathrm{aggregation}}$ [mJ]	$E_{\mathrm{TX},B,\mathrm{aggregation}}$ [mJ]	Savings
02	3	96%	52.40	35.53	32%
04	4	92%	52.40	20.48	61%
06	1	48%	52.40	48.20	8%
09	3	89%	52.40	33.27	37%
13	2	77%	52.40	43.02	18%
16	1	51%	52.40	43.73	16%

**Table 5 sensors-23-04994-t005:** Network parameters used for the network, experimentally deployed on the university campus.

Parameter	Symbol	Value
Route discovery	$T_{\mathrm{route}}$	6 h
Measure interval	$T_{\mathrm{measure}}$	30 min
Minimal aggregation timer	$T_{a,\min}$	0 s
Initial aggregation timer	$T_{a,0}$	12.5 min
Maximal aggregation timer	$T_{a,\max}$	15 min
Aggregation upstep	$T_{a,\mathrm{upstep}}$	1 min
Aggregation downstep	$T_{a,\mathrm{downstep}}$	30 s
TX buffer size		150 B
Payload size		12 B
LoRa spreading factor	SF	7
LoRa bandwidth	BW	500 kHz
Preamble width	$T_{\mathrm{preamble}}$	1.91 s

## Data Availability

All research methods and results are open-source and can be accessed on github.com/dramco/LoRaMultiHop (last accessed on 11 May 2023).

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
