# Peer review of "An Energy-Efficient LoRa Multi-Hop Protocol through Preamble Sampling for Remote Sensing"

_sensors, 2023, doi:10.3390/s23114994_

Round 1

Reviewer 1 Report

the author presents a multi-hop protocol to extend the sensor’s range with consideration of low-power operation.  the efficiency and reliability of the network have been proved.  the manusctipy provides sufficient support for these research. 

the assumptio for the calculation (see line 202,203) could be further elaborated.

Author Response

Dear Reviewer,

Thank you for taking the time to review our paper.

1. "the assumption for the calculation (see line 202,203) could be further elaborated."

We have revisited and clarified the assumptions with regards to the presented battery lifetime calculations. In these calculations, we assume that the node, for which we determine the battery lifetime, serves as an intermediate node that exclusively forwards data. Each incoming message is assumed to be forwarded accordingly. In the revised version of the paper, we have explicitly outlined these changes.

Sincerely,

Guus Leenders

Reviewer 2 Report

The paper is in great shape. One major comment from my end:

How is the LoRa packet constructed in this study? Can the authors please use the LoRa emulator provided in 10.1109/ACCESS.2021.3070976, and how is that different?

no comments

Author Response

Dear Reviewer,

We appreciate the time you took to review our paper.

1. "How is the LoRa packet constructed in this study?"

The LoRa packet is structured as depicted in Figure 3 in our manuscript. The message structure depicted in Figure 6 refers to the LoRa payload in the general LoRa packet structure (from Figure 3). We have clarified this in our updated manuscript. 

2. "Can the authors please use the LoRa emulator provided in 10.1109/ACCESS.2021.3070976, and how is that different?"

Thank you for referring to this article, which introduces a more complex LoRa link simulator. Our Python simulator primarily concentrates on simulating the multi-hop network aspect. We acknowledge that this emulator represents a valuable extension to our research, and thus, we have included it as a potential expansion of our simulator in Section 4.3.

Sincerely,

Guus Leenders

Reviewer 3 Report

Abstract please add the problem statement. The need for energy efficient LoRa multi-hop
protocol
A flowchart to describe the step-by-step process or procedure will help the readers to follow the
study being conducted.
Line 45-46: To support low-power operation, reception of messages in the multi-hop is entirely based on a
sporadic LoRa channel activity detection (CAD) mechanism. please provide reference
Line 220-221: The next Figure mention in text in Figure 10b, jump from Figure 4 in the text.

Minor

Author Response

Dear Reviewer,

We extend our gratitude for dedicating your time to reviewing our paper.

1. "Abstract please add the problem statement. The need for energy efficient LoRa multi-hoprotocol"

We have updated the manuscript to include a problem statement. These changes are highlighted in the updated manuscript.

2. "A flowchart to describe the step-by-step process or procedure will help the readers to follow the study being conducted."

We have added a flowchart to describe our research methodology. This is depicted in Figure 2 of the updated manuscript. 

3. "Line 45-46: To support low-power operation, reception of messages in the multi-hop is entirely based on a sporadic LoRa channel activity detection (CAD) mechanism. please provide reference"

We have rephrased this to make it more clear that our multi-hop network exclusively uses a periodic CAD mechanism to receive messages. We have provided a reference to indicate that this CAD mechanism is indeed more energy efficient. 

4. Line 220-221: The next Figure mention in text in Figure 10b, jump from Figure 4 in the text.

We have clarified this in our updated manuscript. In the battery life calculations we make use of the energy model that is presented only later in our work, as this is related to the embedded implementation of the presented multi-hop network. 

Sincerely,

Guus Leenders

Reviewer 4 Report

Discussed paper is discussing new energy-efficient LoRa multy-hop protocol for remote sensing. Energy-efficiency is the one of the key issues in the development and application of wide sensors network for city IoT or agricultural IoT. Minimization energy consumption gives opportunity to use energy-harvesting technologies and make those sensors network fully autonomous and enhance reliability. Presented in the paper protocol demonstrates reliable energy consumption decrease for tested sensor network, thus it can find wide practical application in the nearest future. Paper can be accepted after minor revision, main comments are:

-          Please expand the introduction to highlight necessity of the development of energy efficient protocols;

-          Please give more information concerning sensors network that was used in the section 5.2.2 (type of sensors, transmitters and etc.).

Author Response

Dear Reviewer,

Thank you for dedicating your time to review our paper.

1. " Please expand the introduction to highlight necessity of the development of energy efficient protocols;"

We have clarified in our updated manuscript as to why low-power operation is a necessity for IoT nodes that are deployed in remote areas. Low-power operation prolongs battery life, reduce maintenance needs, and enables efficient deployment in resource-constrained environments. These changes are highlighted in our updated manuscript. 

2. Please give more information concerning sensors network that was used in the section 5.2.2 (type of sensors, transmitters and etc.).

The IoT sensors used in the network deployed in Section 5.2.2 are in fact the physical sensors that are introduced in Section 3. They do not transmit any useful sensor data, instead they transmit an incrementing counter to accurately measure the resulting packet delivery ratio. We have clarified this in our manuscript and highlighted the applicable changes. 

Sincerely, 

Guus Leenders